# Nanotechnology in Tumor Biomarker Detection: The Potential of Liganded Nanoclusters as Nonlinear Optical Contrast Agents for Molecular Diagnostics of Cancer

**DOI:** 10.3390/cancers13164206

**Published:** 2021-08-21

**Authors:** Guillaume F. Combes, Ana-Marija Vučković, Martina Perić Bakulić, Rodolphe Antoine, Vlasta Bonačić-Koutecky, Katarina Trajković

**Affiliations:** 1Center of Excellence for Science and Technology-Integration of Mediterranean Region (STIM), Faculty of Science, University of Split, 21000 Split, Croatia; guillaume.combes@medils.hr (G.F.C.); anamarija.vuckovic@medils.hr (A.-M.V.); martina@stim.unist.hr (M.P.B.); vbk@cms.hu-berlin.de (V.B.-K.); 2Mediterranean Institute for Life Sciences (MedILS), 21000 Split, Croatia; 3UMR 5306, Centre National de la Recherche Scientifique (CNRS), Institute Lumière Matière, Claude Bernard University Lyon 1, F-69622 Villeurbanne, France; rodolphe.antoine@univ-lyon1.fr; 4Interdisciplinary Center for Advanced Science and Technology (ICAST), University of Split, 21000 Split, Croatia; 5Chemistry Department, Humboldt University of Berlin, 12489 Berlin, Germany

**Keywords:** tumor biomarker, cancer diagnostics, molecular diagnostics of cancer, liganded noble metal quantum nanocluster, precision medicine, bioimaging, contrast agents, nonlinear optics, two-photon-excited fluorescence

## Abstract

**Simple Summary:**

The key factor in preventing premature death from cancer is an early and accurate diagnosis. While common diagnostic procedures are successful in the detection and rough description of a tumor mass, a deeper insight into cancer’s molecular features is needed to optimize the treatment and increase the chances of survival. Nanotechnology can aid the molecular diagnostics of cancers through a design of nanomaterials that can simultaneously recognize specific cancer-associated molecules, so-called tumor biomarkers, and allow for their visualization by different imaging techniques. With a recent explosion in the production of various nanomaterials, the selection of the most suitable nanomaterial for tumor biomarker detection becomes a challenge. In this article, we review recent advances in the molecular diagnostics of cancer using nanotechnology and focus on liganded noble metal quantum nanoclusters, a class of ultrasmall nanomaterials with distinctive structural and optical properties, as tools in tumor biomarker detection.

**Abstract:**

Cancer is one of the leading causes of premature death, and, as such, it can be prevented by developing strategies for early and accurate diagnosis. Cancer diagnostics has evolved from the macroscopic detection of malignant tissues to the fine analysis of tumor biomarkers using personalized medicine approaches. Recently, various nanomaterials have been introduced into the molecular diagnostics of cancer. This has resulted in a number of tumor biomarkers that have been detected in vitro and in vivo using nanodevices and corresponding imaging techniques. Atomically precise ligand-protected noble metal quantum nanoclusters represent an interesting class of nanomaterials with a great potential for the detection of tumor biomarkers. They are characterized by high biocompatibility, low toxicity, and suitability for controlled functionalization with moieties specifically recognizing tumor biomarkers. Their non-linear optical properties are of particular importance as they enable the visualization of nanocluster-labeled tumor biomarkers using non-linear optical techniques such as two-photon-excited fluorescence and second harmonic generation. This article reviews liganded nanoclusters among the different nanomaterials used for molecular cancer diagnosis and the relevance of this new class of nanomaterials as non-linear optical probe and contrast agents.

## 1. Cancer Diagnostics and Nanotechnology

### 1.1. Cancer Diagnostics: From Macroscopic Description to Molecular Diagnostics and Precision Medicine

Cancer is one of the leading causes of premature death globally [1]. In 2020, the GLOBOCAN online database estimated almost 10 million cancer deaths in 185 countries for 36 types of cancer [2]. Furthermore, cancer patients with comorbidities have higher chances of dying from non-cancer-related causes [3], which further increases the overall health and economic impact of cancer. So far, the most efficient strategy to reduce cancer mortality and comorbidity rates and, consequently, the associated burden on the health care system has been the early detection and diagnosis of cancer. However, despite tremendous efforts, the early detection of cancer remains challenging, mainly due to the complex nature of cancer. Moreover, fine features of specific tumors, which can be instructional in designing and customizing therapeutic approaches, often escape standard diagnostic procedures. 

Traditionally, clinicians have been able to reveal the presence of cancer only upon the development of the first symptoms, which usually happens after a tumor mass has reached a substantial size or even after the spreading of metastases. Samples of such tumors can be extracted by tissue biopsies and characterized using standardized histopathological techniques that enable the rough description and categorization of cancer cells. Nevertheless, as biopsies are limited to small amounts of cancer tissue, this procedure provides little information about the overall heterogeneity of cancers. Moreover, such invasive tissue sampling cannot be performed repeatedly as it may inflict complications [4].

Some limitations of tissue biopsies have been overcome by the development of imaging approaches, whereby entire tumors are visualized in situ. The foundations laid more than a century ago by the discovery of X-rays [5] have allowed for the development of techniques such as X-ray computed tomography (CT), positron emission tomography (PET), single-photon emission computed tomography (SPECT), and magnetic resonance imaging (MRI) (Table 1). These methods have an improved diagnostic capacity relative to tissue biopsy as they can provide information about cancer processes, location, and stage [6]. In particular, anatomy-based imaging (CT and MRI) provides information about the location, size, and morphology of the cancer, and function-based imaging (PET and SPECT) map physiological and biological processes within the cancer. Combining anatomy-based imaging with function-based imaging further increases diagnostic power. The introduction of hybrid imaging such as SPECT/CT, PET/CT, optical/CT, and PET/MRI has improved diagnostic accuracy in oncology, but equipment and operational costs account for their slow implementation [7]. Another imaging approach, optical molecular imaging, is holding great promise for cancer diagnosis. Optical imaging (OI) can currently reach wide spatial imaging scales, ranging from cells to organ systems, which renders this technology extremely appealing for medical imaging. Moreover, OI has at its disposal diverse contrast mechanisms (using light absorption and emission methods as well as hybrid OI approaches) for distinguishing normal from pathologic processes and tissues in both small animal and human studies. Typically, complex methods are needed to extract quantitative data from deep tissues.

In the 1970s, with the development of nuclear medicine and the application of radioisotope tracers that are internalized by cancer cells [8], the focus shifted to the molecular events associated with cancer. An example of such an approach is the application of 2-^18^F-fluorodeoxyglucose (FDG), a glucose analog whose accumulation increases in cancer cells relying on glucose metabolism [9,10]. FDG is still extensively used for detection by PET imaging of various cancers, including breast and colorectal cancers, melanomas, and lymphomas. However, FDG labeling has limited specificity since non-cancer cells can also metabolize glucose. The need for increased specificity has prompted a search for new tumor-specific entities, with the goal to develop reliable detection strategies that will aid the early detection of cancer.

The shift towards the molecular profiling of cancer in the late 20th century has paved a road to a precision medicine approach where molecular profiling is combined with large biomedical data sets and used for precision diagnostics, prognostics, and therapeutic strategies in the cancer field [11]. A fundamental goal of precision medicine is to provide effective therapeutic strategies for patients based on their individual cellular, molecular, and biomarker characteristics, along with their unique genetic and environmental factors [12]. Hence, the terms “precision medicine” and “personalized medicine” are often used interchangeably. This approach is driven by technological advances, the interpretation of -omics data, and the development of targeted therapies based on the identification of tumor biomarkers [13,14].

Tumor biomarkers are specific entities produced by cancer cells that can be found in the tumor itself or in the tissues and body fluids of patients. Altered levels of a wide range of entities, such as cells, proteins, peptides, post-translational modifications, metabolites, nucleic acids, and genetic variations, can reveal the presence of cancer in the body as well as help in individualized risk factor assessments, prognostics, and therapy response prediction [15]. Additionally, monitoring patients’ response to treatment can identify molecular alterations occurring during treatment and help navigate the course of the therapy. For instance, genomic alternations such as specific mutations in breast cancer 1 (*BRCA1*) and breast cancer 2 (*BRCA2*) genes are well-established tumor biomarkers used for breast cancer risk assessment, meaning that women bearing these mutations are predisposed to develop breast cancer [16,17]. Moreover, it has been shown that the BRCA status predicts the responsiveness to therapies that interfere with DNA repair machinery, such as cisplatin or olaparib [18,19]; in particular, patients that do not bear these specific mutations are unlikely to respond to such treatments. Hence, unnecessary exposure to toxic therapies can be avoided by genotyping patients. This example highlights the complexity of cancer and the need for the discovery of effective tumor biomarkers that will help in achieving the goals of precision medicine and allow the stratification of patients. 

### 1.2. Methods in Molecular Diagnostics of Cancer

The development of methods for analysis and the monitoring of the tumor biomarker landscape during tumorigenesis is of great importance for accurate cancer diagnostics and the design of personalized therapeutic options. Indeed, the introduction of genomics, epigenetics, proteomics, metabolomics, informatics, and imaging techniques has greatly improved our understanding of cancer’s molecular features and allowed for improved survival prospects for patients by matching tumor characteristics with complementary therapy [20]. These methods are applicable to the analyses of tumor tissue biopsies as well as liquid biopsies, which contain biomarkers released by the tumor into bodily fluids, such as circulating tumor nucleic acids and circulating tumor cells (CTC) [21,22].

At present, the characterization of the cancer genome is performed by polymerase chain reaction (PCR) assays, high-throughput DNA microarrays, or fluorescence in situ hybridization (FISH) [23,24]. These techniques enabled the detection of short tandem repeats [25], loss of heterozygosity [26], alterations in DNA methylation status, and various mutations. For instance, sequencing for genetic alterations in human epidermal growth factor receptor (*EGFR*), human epidermal growth factor receptor 2 (*HER2*), vascular endothelial growth factor (*VEGFR*), rearranged during transfection (*RET*) tyrosine kinase, mesenchymal–epithelial transition factor (*MET*), mitogen-activated protein kinase (*MEK*), anaplastic lymphocyte kinase (*ALK*), and ROS proto-oncogene 1 (*ROS1*) has allowed for precision therapy of non-small cell lung cancer, liver cancer, breast cancer, and renal cell carcinoma based on matching the individual’s cancer mutations with a specific tyrosine kinase inhibitor as a treatment of choice [27,28].

Specific detection and identification of protein biomarkers in clinics have been mainly accomplished by various immunological techniques based on labeling sensors, where the final result is proportional to the amount of label bound to the targeted molecule [29,30]. The enzyme-linked immunosorbent assay (ELISA) is widely used for protein detection and quantification and is based on the use of specific binding surfaces decorated with antibodies that capture protein biomarkers from various body fluids. Protein biomarkers are then revealed by enzyme-labeled antibodies [31] that produce a fluorescent signal/color that corresponds to the amount of biomarker immobilized by the capturing antibody. ELISA displays high sensitivity–for instance, it can already detect the prostate-specific antigen (PSA), which is used for prostate cancer screening, at a concentration of 100 pg/mL [32]. However, the use of a single cancer protein biomarker is not sufficient for accurate cancer detection, as PSA can also be detected in patients with benign prostatic hyperplasia [33]. Therefore, to improve diagnostic accuracy, various multiplex immunoassays that combine the antibody-labeled detection of different protein biomarkers are needed. An example of a multiplex immunoassay is the FDA-approved OVA1^®^ test that combines the detection of multiple serum-derived protein cancer antigen 125 (CA 125), transferrin, apolipoprotein A-I (APOA1), β2-microglobulin, and transthyretin with software calculations to predict ovarian malignancy [34]. In the second generation OVA2^®^ (Overa) test, the latter two-protein biomarkers are replaced with human epididymis protein 4 (HE4) and follicle-stimulating hormone (FSH), having improved accuracy and reduced false results [35,36]. Due to their high sensitivity, immunological techniques are also used for the detection of CTC. For example, the CellSearch^®^ system uses nanoparticles labeled with antibodies that target the epithelial cell adhesion molecule (EpCAM) to separate CTC from other cells present in the blood of patients with metastatic colorectal, breast, or prostate cancer. Immunomagnetic separation is followed by the immunocytological detection of various cytokeratins and leukocyte common antigen CD45 as well as the analysis of the cellular nuclei using a fluorescence microscope.

The recently developed CancerSEEK test [37] combines the multiplex measurement of genetic alterations in ctDNA with the measurement of protein levels of CA 125, cancer antigen 19-9 (CA 19-9), carcinoembryonic antigen (CEA), osteopontin (OPN), prolactin (PRL), myeloperoxidase (MPO), tissue inhibitor of metalloproteinases 1 (TIMP1), and hepatocyte growth factor (HGF) in blood samples. Through the measurement of ctDNA and protein biomarkers, the CancerSEEK tumor biomarker detection test displays 69–98% sensitivity in detecting liver, ovarian, stomach, esophageal, and pancreatic cancer.

### 1.3. Metallic Nanomaterials in Detection of Tumor Biomarkers

#### 1.3.1. Nanotechnology in Cancer Diagnostics: From Large Nanoparticles (NPs) to Nanoclusters (NCs)

Despite the progress in understanding cancer biology, the diagnostics of cancer still faces many challenges [38]. Currently used clinical procedures are often invasive and unpleasant and have limited potential to detect specific molecular events [39]. Hence, there is a great need to improve cancer diagnostics by developing tools that might complement the existing clinical approaches or novel, non-invasive alternatives with enhanced specificity and sensitivity.

In the past few decades, nano-sized devices have undergone rapid development due to their immense potential in biomedical applications [40]. Their ability to act on a cellular or subcellular level has prompted an enormous number of studies exploring a large panel of organic (liposomes, micelles, biopolymeric NPs, dendrimers) and inorganic (metal NPs, quantum dots, carbon nanotubes, nanoshells, nanocrystals) nanomaterials. The versatility of nanomaterials offers room for the optimization of their stability, selectivity, biological targeting properties, and detection. Some nanomaterials have already been introduced to (pre-)clinical cancer diagnostics, as is the case with iron oxide nanoparticles, which are used as a contrast agent for MRI imaging [41,42]. Importantly, suitable surface modification of nanomaterials enables their selective binding to a specific biological target, thus opening opportunities for the expansion of use of metal-based nanomaterials in various therapeutic and targeting systems [43].

Over the past few decades, various metal-based nanomaterials, composed of gold, silver, or copper, have been synthesized for molecular imaging and drug delivery purposes. Among all metallic nanomaterials, gold nanomaterials exhibit superior stability and optical properties and generally have lower toxicity, which makes them good candidates for potential clinical applications in diagnostics and therapy [44] and, in particular, in personalized medicine approaches.

Generally, metallic nanomaterials can be classified into three size groups: large NPs, small NPs, and quantum nanoclusters (NCs) [45]. Large NPs are larger than 50 nm and can be seen as small metal spheres [46], the optical properties of which depend on their volume and dielectric constant. For the second group of small NPs, with the size of 2–50 nm, the dielectric function becomes size-dependent, and the optical response is of a plasmonic nature. Finally, NCs are drastically different due to their ultrasmall size (<2 nm) and molecule-like properties, characterized by quantum discrete states. Of note, nanomaterial solutions are often heterogeneous in terms of individual particle sizes as it is difficult to achieve uniformity during synthesis, and, moreover, the agglomeration of individual particles can occur [47]. Average uniformity is expressed as a polydispersity index (PDI)—the square of the standard deviation of the particle diameter distribution divided by the mean particle diameter. PDI reflects the quality of the nanomaterial solution, and it is relevant to their clinical application.

#### 1.3.2. Functionalization of Nanomaterials

The use of nanomaterials for cancer diagnostics is based on the functionalization of their surface by various ligands that serve to (1) maintain their physicochemical properties in vitro and in the biological environment (structural functionalization) and (2) provide them with specific target-recognition properties (specializing functionalization) [48,49]. Structural functionalization is used to protect the metallic nanomaterial from harsh conditions (such as suboptimal pH) as well as to reduce its toxicity and improve bioavailability. Compounds containing sulfur, such as cysteine (Cys), glutathione (GSH), mercaptopropionic acid (MPA), bidentate dihydrolipoic acid (DHLA), and thiolated polyethylene glycol (PEG), are the most common ligands used for structural functionalization.

Specializing functionalization adds various recognition moieties to metallic nanomaterials, which makes them competent to efficiently detect specific molecular targets and deliver drugs to desired destinations within the biological systems [50]. Those recognition moieties include peptides, antibodies, proteins, aptamers, small molecules, or polymers, and they enable the binding of functionalized nanomaterials to a specific tumor entity. The resulting complex between a tumor entity and nanomaterial can then be detected by suitable imaging techniques.

The addition of a recognition moiety can be achieved by a ligand exchange strategy based on the replacement of a structural ligand with a specific biomolecule or by a conjugation strategy that allows the binding of the molecule of interest to the structural ligands on the metallic nanomaterial. The ligand substitution induces size and structure transformation, and, generally, this approach is more difficult to apply on small NPs than on NCs [51].

Conjugation is usually based on the activation of the carboxylic acid of the protective ligand with N-(3-dimethylaminopropyl)-N*-ethylcarbodiimide hydrochloride (EDC), which allows the formation of amide bonds between carboxyl groups of the ligand with amine residues on the protein surface. One such example is the work of Jazayeri et al., where a PEGylated gold nanoparticle was conjugated with an anti-PSA antibody using an EDC/N-hydroxylsuccinimide (NHS) linker [52]. The conjugation of antibodies such as anti-EGFR [53] and anti-Her2 [54,55,56] is of particular interest as they are already in clinical use for cancer detection and effective tumor targeting [57]. Over the recent years, the addition of a variety of ligands by conjugation [58,59,60,61,62] has become the method of choice as it offers tremendous opportunities in molecular recognition and targeting.

#### 1.3.3. Tumor Biomarker Detection by Metallic Nanomaterials 

In the past couple of decades, a number of metallic nanomaterials targeting various tumor biomarkers have been developed (reviewed in [63,64,65]). Here, we summarize recent examples of such nanomaterials, mostly metal NPs and NCs (Table 2). Many of them recognize receptors overexpressed on the surface of malignant cells, including the folate receptor, EGFR, HER2, GLUT, GRPR, and CCR5. Selection of the plasma-membrane-located receptors as targets is a reasonable strategy in the design of nanomaterials as it increases the probability that the nanomaterial will encounter the target and get internalized into the cells. In such instances, specializing ligands are either the natural ligands (e.g., folic acid for the folate receptor or glucose for GLUT) or peptides, aptamers, and antibodies designed to specifically target the receptors.

Metallic nanomaterials bound to their molecular targets can be detected using multiple techniques, the choice of which depends on the properties of the nanomaterial and the compatibility of these techniques with the type of nanomaterial-labeled biosample—cell line, tissue, liquid biopsy, or the entire organism. While some nanomaterials, such as AuNCs, are intrinsically luminescent in the range from visible to near-infrared (NIR) light [66,67], others require adaptation to the desired detection method by functionalization. In cell culture samples, such nanomaterials can be detected using fluorescence microscopy or less common methods, such as inductively coupled plasma mass spectrometry (ICP-MS), inductively coupled plasma atomic emission spectroscopy (ICP-AES), auto-metallography, and surface-enhanced Raman spectroscopy (SERS) [56,68,69]. Nanomaterial-labeled liquid biopsies (human serum and urine) allow the application of other methods such as fluorescence spectroscopy, colorimetric detection, and surface plasmon resonance (SPR) biosensors [70,71]. Finally, nanomaterials can be detected in vivo, e.g., in xenografted mice using MRI, nuclear imaging methods (SPECT, PET/CT, autoradiography), NIR fluorescence imaging systems, or photoacoustic imaging. For MRI, it is required for the nanomaterials to include or to be coated with a heavy metal such as iron oxide or gadolinium [72,73]. On the other hand, nuclear imaging methods require the labeling of the nanomaterials with radioisotopes [74]. Among these techniques, relatively novel photoacoustic imaging has been viewed as one of the most promising imaging techniques due to a relatively large imaging depth (up to 1 cm) [75]. This non-invasive imaging modality uses the optical properties of the nanomaterial and is compatible with both AuNPs and AuNCs [76,77,78].

Despite the huge variety of different nanomaterials developed for cancer diagnostics, only a small number have progressed to clinical trials [64]. Currently, there are 27 nanoparticles under clinical investigation for cancer diagnostics registered in clinicaltrials.gov. Despite their potential for biomedical applications, there are still biological (e.g., biodistribution, metabolism, pharmacokinetics), technological, safety, and regulatory challenges that need to be thoroughly investigated [65].

**Table 2 cancers-13-04206-t002:** Examples of cancer biomarker detection by different types of metallic nanomaterials.

Biomarker Detection			Functionalization		Nanoparticle/Nanocluster			
Method	Biomarker	Model	Type of Ligand	Ligand	Composition	Dimension (nm)	PDI	Ref.
**Fluorescence** **microscopy**	Folate receptor	Lung, breast cancer cell lines	Small molecule	Folic Acid	Au_25_-BSA	~1/~8 (aggregates)	-	[58]
	PrP^c^	Colorectal cancer cell line	Oligonucleotide	PrPC aptamer	AuNPs	13/20 (NPs/NPs + PrPc)	-	[79]
	EGFR	Lung cancer cell lines	Protein	EGFR-specific scFv	Fe_3_O_4_/AuNPs	30/76.3 (NPs/NPs + scFv)	-	[80]
	HER2	Breast cancer cell lines	Peptide	Anti-HER2 peptide	Fe_3_O_4_/AuNPs-Cy5.5	~12	0.08	[81]
	-	In vitro blood–brain barrier model	Amino acid	L-Dopa	AuNCs-SG	1.4	-	[82]
	CD44 receptor	Lung, breast cancer cell lines	Glycosaminoglycans	Hyaluronic acid	Au-SG-Graphene oxide	2	-	[83]
	Glutathione	Cervical, liver, colon cancer cell lines	Protein	Transferrin	AuNCs	4.72 ± 0.5	-	[84]
	Methionine level	Lung, liver, breast cancer cell lines	Amino acid	Methionine	Au-MET-MPA	5.6	0.118	[85]
	PD-L1	Colon cancer cell line	Antibody	anti-PD-L1 Ab	AuNPs-PEG	40.0 ± 3.1	-	[86]
	GLUT1	Breast cancer cell line	Small molecule	Glucose	AuNPs	47 (DLS)	0.15	[87]
	Neuron	In vitro blood–brain barrier model Mouse	Extracellular vesicle	Exosome	AuNPs	105 ± 10.1 (DLS)	0.430 ± 0.06	[88]
	Neuron	Rat	Protein	WGA-HRP	AuNPs-MSA	5.2 ± 1.3	-	[89]
	Folate receptor	Ovarian cancer cell line	Small molecule	Folic acid	AuNCs-BSA	25 ± 12 (DLS)	-	[90]
	Thyroid	Thyroid carcinoma cancer cell line	Chemical element	^127^Iodine	AuNCs-BSA-I^127^	6.4 (DLS)	-	[68]
	Calreticulin	Colon, breast cancer cell lines	Antibody	Anti-calreticulin Ab	AuNCs-MSA	2	-	[91]
	HER2	Breast cancer cell lines	Oligonucleotide	Affibody-DNA	AuNPs	18.5 ± 1.1/31.7 ± 1.3 (NPs/NPs + affibody) (DLS)	-	[92]
	HER2	Stomach cancer cell lines	Antibody	Tmab	AuNPs	85.39 ± 0.68 (DLS)	-	[93]
	PSMA	Prostate cancer cell lines	Peptide	PSMA-1	AuNPs-PEG-Pc4	5.5 ± 0.4 (AuNPs-PEG)	-	[94]
	Folate receptor	Bone, cervical, lung cancer cell lines	Small molecule	Folic acid	Au_22_SG_18_	1.4	-	[95]
	Leukemia cells	Leukemia cancer cell line	Oligonucleotide	KH1C12 aptamer	Fe_3_O_4_/AuNPs	26	-	[72]
**Colorimetric**	MMP9	Colon cancer mice urine	Peptide	Protease-cleavable peptide	AuNCs-SG	~1.5	-	[96]
	Citrate	Prostate cancer—Human urine			AuNCs-Cys	4–6	-	[97]
	hCG	Testicular cancer—Human urine Serum	Peptide	hCG-specific peptide aptamer	AuNPs	13	-	[98]
**Fluorescence spectroscopy**	Alkaline phosphatase	Human serum	Small molecule	Pyridoxal phosphate	AuNCs-BSA	1.95	-	[99]
	Glutathione	Glutathione	Small molecule	Folic acid	AuNCs-BSA-rGO	<2	-	[100]
**ICP-MS**	MMP9	Colon cancer—Human urine	Peptide	Protease-cleavable peptide	AuNCs-SG	~1.5	-	[96]
	HER2	Breast cancer cell lines	Antibody Peptide	Trastuzumab HIV-TAT cell-penetrating peptide	AuNPs-PEG	87.35 ± 0.41 (DLS)	0.17	[56]
	GLUT	Xenografted breast cancer mice	Small molecule	Glucose	AuNPs	47 (DLS)	-	[87]
**ICP-AES**	EGFR	Epidermoid carcinoma cancer cell line	Antibodies	VHH 122 Ab Cetuximab	AuNPs-PEG	28/42/45/63 (DLS) (NPs/+PEG/+VHH/+cet)	0.22/0.30/0.31/0.24	[101]
	Folate receptor HER2	Breast cancer cell line	Small molecule Antibody	Folic Acid Herceptin	AuNCs-BSA	4.2/9.8 (NCs/NCs + FA + HER)	-	[102]
	-	In vitro blood–brain barrier model	Amino acid	L-Dopa	AuNCs-SG	1.4	-	[82]
	Thyroid	Patient-derived xenografted thyroid cancer mice	Chemical element	^127^Iodine	AuNCs-BSA-I^127^	6.4 (DLS)	-	[68]
	EGFR	Lung and colorectal cancer cell line Xenografted colorectal cancer mice	Antibody	Cetuximab	AuNPs-PEG	78.3 ± 0.7 (DLS)	-	[103]
	CA 19.9 antigen	Pancreatic cancer cell lines Xenografted pancreatic cancer mice	Antibody	5B1 antibody	AuNPs-Zr^89^	34.86 (DLS)	0.27	[104]
**SPR** **biosensor**	CEA	Colon cancer—Human plasma	Antibody	Anti-CEA Ab	AuNPs	30 ± 6 (NPs)	-	[70,71]
**LSPR** **biosensor**	PSA	Prostate cancer—Protein (PSA)	Antibody	Anti-PSA Ab	AuNPs-PEG	25	-	[52]
**SERS**	CD19	Leukemia cancer cell line	Antibody	Anti-CD19 Ab	AuNPs-PEG-MGITC	60 (DLS)	-	[105]
	CEA	Breast and lung cancer cell lines	Antibody	Anti-CEA Ab	AuNPs-Fe_3_O_4_-ATP-4	~20 (DLS)	-	[106]
	MCSP, MCAM, ErbB3, LNGFR	Melanoma, breast cancer cell lines Human plasma	Antibodies	Anti-MCSP, anti-MCAM, anti-ErbB3, anti-LNGFR, Abs	AuNPs-MBA, AuNPs-BA-TFM, AuNPs-DNTB, AuNPs-MPY.	60 (NPs alone)	-	[69]
**MRI**	EGFR	Xenografted lung cancer mice	Protein	EGFR-specific scFv	Fe_3_O_4_/AuNPs	30 (NPs alone) 76.3 (NPs + scFv)	-	[80]
	Leukemia cells	Leukemia cancer cell line	Oligonucleotide	KH1C12 aptamer	Fe_3_O_4_/AuNPs	26	-	[72]
	Nucleolin	Breast cancer cell line	Oligonucleotide	AS1411 aptamer	AuNPs-DO3A-Gd(III)	3.4 ± 0.6	-	[73]
**NIR** **fluorescence** **imaging** **system**	HER2	Xenografted breast cancer mice	Peptide	Anti-HER2 peptide	Fe_3_O_4_/AuNPs-Cy5.5	~12	0.08	[81]
	EGFR	Xenografted lung cancer mice	Protein	EGFR-specific scFv	Fe3O4/AuNPs	30 (NPs alone) 76.3 (NPs + scFv)	-	[80]
	-	Mouse	Amino acid	L-Dopa	AuNCs-SG	1.4	-	[82]
	Methionine level	Xenografted breast and lung cancer mice	Amino acid	Methionine	AuNCs-MET-MPA	5.6	0.118	[85]
	-	Mouse	-	-	AuNPs-SG-Au^198^	2.6 ± 0.3	-	[107]
	PSMA	Xenografted prostate cancer mice	Peptide	PSMA-1	AuNPs-PEG-Pc4	5.5 ± 0.4 (AuNPs-PEG)	-	[94]
	Thyroid	Patient-derived xenografted thyroid cancer mice	Chemical element	^127^Iodine	AuNCs-BSA-I^127^	6.4 (DLS)	-	[68]
	Folate receptor	Mouse	Small molecule	Folic acid	Au_22_SG_18_	1.4	-	[95]
**SPECT**	-	Mouse	-	-	AuNPs-SG-Au^198^	2.6 ± 0.3	-	[107]
	CCR5	Xenografted breast cancer mice	Peptide	D-Ala1-peptide T-amide	^199^AuNPs-PEG	5	-	[108]
**PET/CT**	LHRH receptor	Xenografted lung cancer mice	Peptide	LHRH peptide	AuNCs-HSA-I^124^	2	-	[74]
	Thyroid	Patient-derived xenografted thyroid cancer mice	Chemical element	^127^Iodine	AuNCs-BSA-I^127^	6.4 (DLS)		[68]
	EGFR	Xenografted epidermoid carcinoma cancer mice	Antibodies	VHH 122 Ab Cetuximab	AuNPs-PEG	28/42/45/63(DLS) (NPs/+PEG/+VHH/+cet)	0.22/0.30/ 0.31/0.24	[101]
	CA 19.9 antigen	Pancreatic cancer cell lines Xenografted pancreatic cancer mice	Antibody	5B1 antibody	AuNPs-Zr^89^	34.86 (DLS)	0.27	[104]
**-**	EGFR	Colon cancer cell lines	Antibody	Anti-EGFR Ab	AuNPs	14.9 ± 1.23	-	[109]
**Auto-radiography**	CCR5	Xenografted breast cancer mice	Peptide	D-Ala1-peptide T-amide	^199^AuNPs-PEG	5	-	[108]
**Auto-metallography**	EGFR	Glioblastoma, fibrosarcoma cancer cell lines	Antibody	Cetuximab	AuNPs	2.4 ± 0.28	-	[110]
**Photoacoustic** **imaging**	EGFR	Epidermoid carcinoma cancer cell line	Antibody	Anti-EGFR Ab	AuNPs	5	-	[76]
	GRPR	Prostate cancer cell lines Xenografted prostate cancer mice	Peptides	GRPR-targeting peptides	Au nanorod	8 ± 2 nm (W)/49 ± 8 nm (L)	-	[77]

## 2. Potential of Quantum Nanoclusters as Non-Linear Optical Probes in Molecular Diagnostics of Cancer

Given the variety of options for nanomaterials as well as for the specializing and structural ligands, the choice of an optimal nanodevice in designing strategies for tumor biomarker detection presents a challenge. In this section, we elaborate on opportunities provided by optical molecular imaging as a technique for biomarker detection and gold NCs as non-linear optical (NLO) probes. NCs are particularly interesting due to their distinctive physicochemical properties. However, these unique properties of the NCs are often poorly recognized in the literature, promoting the perception of the NCs as merely a smaller version of NPs. Here, we emphasize the key physicochemical features of the NCs, in particular those features responsible for their biocompatible optical properties that might be relevant for their application in cancer diagnostics. 

### 2.1. Non-Linear Optical Techniques and Nanomaterials

Optical molecular imaging is holding great promise in cancer diagnosis. However, a critical issue for photo-induced imaging is the capability of light to penetrate tissues. Since tissues such as blood, fat, and skin inherently interact with any incident light, leading to light absorption and/or scattering, it is essential for OI to operate at wavelengths where light attenuation is minimal. The first NIR (NIR-I) window, between 700 and 900 nm, and, more interestingly, the second (NIR-II) window fulfill this requirement for biological imaging (Figure 1a) [111]. Thus, any contrast agents with small optical gaps and, hence, with absorption (and possibly emission) in the NIR-II (1000–1400 nm) are highly desired (Figure 1b). For this purpose, a plethora of exogenous contrast agents has been developed, including inorganic imaging contrast agents and organic probes [111]. However, a major challenge concerns the elucidation of the biocompatibility, pharmacokinetics, and long-term toxicological profiles of such NIR-II contrast agents. 

As an alternative to using chromophores with small optical gaps, NLO techniques such as second harmonic generation (SHG) and two-photon-excited fluorescence (TPEF) can be exploited (Figure 1c) [114]. SHG is a phenomenon whereby two photons with the same frequency get combined during interaction with a non-linear material, generating a new photon with twice the energy of the initial photons while conserving the coherence of interaction. TPEF is based on simultaneous excitation with two photons, followed by the emission of light with shorter wavelengths, and it is an incoherent phenomenon involving radiative absorption and re-emission. In both cases, the photon energy can match the second (NIR-II) window for biological imaging due to the inherent multi-photon process. Thanks to the complementary information that they can provide, as well as the enhanced contrasts and improved resolution, SHG and TPEF have gained overwhelming popularity among biologists and have emerged as promising tools in the field of pre-clinical and clinical cancer research.

In particular, TPEF imaging has become popular in tissue imaging due to its advantages of longer wavelength excitation (>1000 nm), which minimizes auto-fluorescence and bleaching and allows for a higher 3D resolution and deep tissue imaging in comparison to one-photon fluorescent microscopy [115]. Moreover, a study performed by Lianhuang Li et al. showed that TPEF combined with SHG helped in identifying early gastric cancer by assessing cell size and collagen alternations without using exogenous contrast agents [116]. Interestingly, the sensitivity of such NLO techniques can be dramatically enhanced using exogenous contrast agents with NLO properties. Some organic dyes have already been applied in multi-photon imaging, but their performance has been hampered due to their rapid photo-bleaching and limited two-photon absorption (TPA) cross-sections [117]. Quantum dots might be a superior option for multi-photon imaging as they display large two-photon absorption cross-sections [118], but their strong cytotoxicity and photon-blinking behavior [119] limit their applicability. Designing highly efficient second-order NLO-phores is largely a matter of the fine combination of a high density of delocalized electrons in a symmetrical or unsymmetrical environment. Recent advances in the field of nanotechnology have allowed for the development of nanostructures that display higher diffusion through tissues along with high TPA cross-sections [113,120]. In particular, as discussed below, NLO characteristics of liganded noble metal NCs and the possibility of their functionalization with specific recognition moieties hold promise for the integration of such NCs with tools of precision medicine that might help in early cancer detection as well as the stratification of patients and the development of treatment options.

### 2.2. Quantum Nanoclusters—General Features

NCs are an extremely appealing family of nanomaterials, in particular for bio-imaging applications. NCs are characterized by a small number of metal atoms (between a few and hundreds of atoms) and by molecular-like discrete states for which strong fluorescence might occur. The connection between their structural and optical properties arises in the size regime, in which “each atom counts”, meaning that the removal or addition of a single metal atom can substantially change structural and optical properties. However, NCs have to be stabilized and protected from the environment in order to prevent their degradation or aggregation. The protection of noble metal NCs from photo-dissociation by inorganic matrices or solid gas has been used since 1987 [121]. Dickson et al. first reported in 2002 [122] the role of organic scaffolds, e.g., DNA oligomers, in the synthesis of silver NCs as both metal cluster protection and the key ingredient for enhancing NC emission in the visible to NIR regime. This dual role of ligands is quite general and has found an application in boosting the NLO properties of liganded gold and silver NCs. 

A large variety of ligands, including amino acids, peptides, proteins, small organic molecules, polymers, DNA, and dendrimers, can be used for the protection of NCs. Figure 2 illustrates different ways to form and stabilize NCs in solution [123,124]. Various ligand-engineering strategies have been developed to enhance the emissive properties of NCs [67]. Among the large variety of ligand families, thiolated molecules are particularly suitable due to strong Au-S binding. Additionally, thiols can be found in organic ligands but also in natural biomolecules such as peptides and proteins. AuNCs with an appropriate choice of thiol-containing ligands present with extremely good stability and biocompatibility and attractive emissive properties, i.e., strong emitted fluorescence. These properties authorize in vitro and in vivo detection of thiolated NCs by multimodal imaging techniques. The wavelength of NC luminescence can be tuned from near-ultraviolet (UV) to the IR region, allowing their detection by X-ray CT, OI, MRI, or photoacoustic imaging. Importantly, NCs demonstrate higher brightness and photo-stability in comparison to organics dyes, which are prone to photobleaching [125].

The ligand selection is of key importance not only for optimizing the optical properties of the NCs but also for developing their targeting competency. NCs can be functionalized during the initial synthesis by the addition of selected target-binding molecules directly on the surface of NCs, e.g., via covalent attachment (post-functionalization) or by ligand exchange after liganded NC synthesis if the target molecule of interest has a thiol group (see Section 2.4.3). Finally, sufficient biocompatibility with minimal toxicity is required to qualify any liganded NC as a good candidate for biomedical imaging.

### 2.3. Biological Properties of AuNCs 

#### 2.3.1. Cellular Uptake—Internalization Mechanisms and Cytotoxicity

The cellular uptake of AuNCs has been studied in multiple cellular systems [126]. It has been shown that the internalization of NCs is energy-dependent [127,128] and relies on multiple endocytic mechanisms such as clathrin-mediated endocytosis and micropinocytosis. The caveolin-mediated pathway is also involved, albeit to a lesser extent. Following their uptake, AuNCs were ultimately transferred to the lysosomes and were not able to reach the nucleus even after 24 h of incubation with the cells [127]. The time dimension of AuNC uptake was studied in BaF3 cells using an AuNC biofunctionalized with an aptamer to target the IL6 receptor [129]. This study has shown that the NCs were bound to the cellular membrane after 10 min of incubation with the cells and were internalized into the cells after an additional 10 min. The cellular uptake of AuNCs has also been demonstrated in an in vitro model of the blood–brain barrier (BBB) using an AuNC functionalized with l-3,4-dihydroxyphenylalanine (L-Dopa) to target the brain and cross BBB [82].

The nature of the ligands may affect the cellular uptake of AuNCs. For instance, zwitterionic ligands seem to be more supportive of AuNC internalization than the PEGylated ones in human-derived monocyte dendritic cells [130]. Similarly, MPA-liganded AuNCs were taken up more efficiently than GSH-AuNCs in the normal human colon mucosal epithelial cell line NCM460 [131].

Generally, the cellular uptake of nanomaterials depends on their size and cell type. Controversial results exist about the efficiency of AuNC uptake relative to other nanomaterials. For instance, the internalization of NCs is relatively low compared to larger particles such as quantum dots [132] in the reticuloendothelial system (RES)—phagocytic cells that clear the circulation and tissues from particles and soluble substances. However, in human dendritic cells, AuNCs are internalized via the endocytic pathway more efficiently than the larger AuNPs.

Importantly, when internalized, AuNCs might cause perturbations in the cellular environment. Recently, a study on human primary astrocytes demonstrated that AuNCs were not inert within the cells. Even though no significant cell loss has been observed for AuNC concentrations below 10 µM, alterations were detected in the organellar and redox-responsive transcription factor homeostasis [133]. These effects may also depend on the nature of the ligands. For instance, the type of ligand may determine the type of immune response in dendritic cells [130] as well as the state of intracellular redox signaling [131].

#### 2.3.2. Biodistribution 

Nanomaterials can reach and accumulate in tumors via passive and active targeting. While active targeting depends on specific interactions between nanomaterials and tumors (see Section 1.3.3), passive tumor targeting by nanomaterials precedes the active targeting and is essential to create an opportunity for the occurrence of specific interactions. Indeed, passive and preferential targeting of the tumors by both NPs and NCs have been observed, and this phenomenon has been named the enhanced permeability and retention effect (EPR) [132,134]. EPR can be explained by the presence of pores with sizes of up to 2000 nm within tumors [135]. These pores represent inter-endothelial gaps formed during angiogenesis in the tumors, and they allow NPs to accumulate in cancer tissues at higher concentrations than in normal tissues. Thus, this passive accumulation of NPs in tumors takes advantage of the pathophysiological properties of the tumor tissue. Despite being the foundation of tumor-targeted drug delivery and the NP accumulation principle, the EPR effect in patients has been recently questioned [136,137]; the mechanism of entry of NPs into solid tumors appears to be more intricate than considered earlier [138]. Either way, passive tumor targeting by nanomaterials has certain disadvantages, such as arbitrary targeting, inefficient dispersion of the NPs, and variability among different tumor types and different patients [137]. Interestingly, the EPR effect is generally more pronounced in animal models than in cancer patients, which hampers understanding of the NC biodistribution and translation to clinic of the results obtained in animal studies. 

The biodistribution of NPs and NCs is affected by their interaction with the environment, and this interaction differs for NPs and NCs. Upon administration, NPs are rapidly exposed to protein-rich biological fluids. These proteins interact with the NPs and form a protein corona on their surface [139]. Such protein coronas affect the size and charge of the NPs [140] as well as their stability, dispersibility, pharmacokinetics, and toxicity profiles [141]. Ultimately, the biodistribution of decorated NPs is altered, and they may even get recognized by the immune system (RES). It has been demonstrated that the protein corona promotes the cellular uptake of the NPs by the immune cells of the RES. To avoid recognition by the immune system, PEGylation of the NPs was introduced, which resulted in an increase in the blood circulation of the NPs as a side effect [142].

Currently, little is known about the nano–bio interactions of NCs, and it is still unclear how they interact with proteins from the biological environment. Interestingly, Yin et al. showed that the conventional protein corona model in DHLA-liganded AuNCs does not apply, and they coined the term “protein complex” for proteins bound to NCs [143]. More studies are required to characterize the NC–protein interaction and its impact on the biodistribution, cellular uptake, and cytotoxicity of the NCs. 

The fine-tuning of nanomaterial biodistribution within tissues and cells can be achieved by active targeting, as described in Section 1.3.3. A variety of ligands can be used for this purpose, and they are usually small molecules that specifically interact with receptors overexpressed at the surface of the tumor cells. Antibodies, aptamers, or peptides are often used since they target proteins on the cell surface [74] and increase the probability of endocytosis of the nanomaterials by tumor cells.

#### 2.3.3. Toxicity and Clearance 

Clinical application of the nanomaterials depends on their toxicity and clearance from the body. Regarding the overall toxicity of nanomaterials, it is important to consider their toxic effects both in vitro and in vivo. High toxicity in vitro (in the cell culture models) can be counteracted in vivo by efficient clearance and vice versa. 

The size, shape, surface properties, and chemical composition of the nanomaterials are critical determinants of their toxicity and clearance [144]. However, the results of various toxicological studies are controversial, making it difficult to derive straightforward conclusions. While some researchers have shown that nanomaterials smaller than 5 nm (which include the NCs) are more toxic than the larger ones, both in vitro [145,146,147] and in vivo, in the zebrafish model [148], others have reported the opposite results in vivo [149,150,151].

Clearance mechanisms have been well described for NPs [152]. Three systems are involved in the clearance of NPs. The first one is the RES, where macrophages phagocytose large NPs (>6 nm), leading to extended retention (up to 6 months) of the partially digested NPs in the body. The second clearance pathway is hepatobiliary excretion [153]. This pathway is also utilized by NPs larger than 6 nm, and their retention, in this case, lasts for up to a couple of weeks. The third elimination route is renal excretion, where the glomerular capillary walls act as a filter for NPs smaller than 6 nm [154]. Renal clearance is often preferred because of the fast and efficient removal of NPs (hours to days), especially non-degradable noble metal NPs [152]. With their small size (<2nm), Au liganded NCs are cleared in vivo through the renal system, which makes them excellent candidates for clinical applications. 

Given that the biological safety of nanomaterials depends on many intrinsic and extrinsic factors, including their biological environment, toxicity must be evaluated for each specific NC.

### 2.4. Structure, Optical Properties, and Functionalization of Quantum Nanoclusters 

#### 2.4.1. Structure of Quantum Nanoclusters 

To synthesize AuNCs containing only several atoms, an appropriate combination of parameters such as temperature, stabilizers, reduction method, and the initial ratio of metal salt to stabilizer is needed. The atomic precision and molecular purity of AuNC can be reached using size-focused methodology [155,156,157]. In addition, the use of ligands is crucial not only for the stabilization of AuNCs but also for tailoring their fluorescent properties. Suitable ligands with electron-rich atomic groups can enhance the fluorescence of AuNCs due to the charge transfer between the ligands and the metal core [158]. Thiols are commonly employed as ligands for noble metal NCs because of the strong affinity of sulfur to noble metals, especially to gold. Moreover, thiol-containing molecules are good stabilizers for AuNCs. Among thiols, GSH has played a key role in the production of AuNCs [159,160,161].

A distinctive feature of atomically precise ligand-protected noble metal NCs is the connection between their structure and their spectroscopic properties. The properties of such NCs can be determined using experimental and theoretical approaches, which provide complementary information. Different techniques, such as X-ray crystallography, mass spectrometry, X-ray powder diffraction (XRPD), and nuclear magnetic resonance (NMR) spectroscopy, have been used for the characterization of AuNCs [45,123,162,163]. The structure of some NCs has been resolved experimentally by means of a single crystal X-ray diffraction [123].

Theoretical approaches include density functional theory (DFT) and time-dependent density functional theory (TDDFT), where a basic variable, the many-body wavefunction, is replaced by a density function. The TDDFT approach allows the calculation of the photoabsorption spectra for relatively large systems after the structural properties have been determined by DFT. Despite significant approximations, these theoretical approaches are useful for predictions of structural and optical properties of both protected and unprotected NCs [164].

The combined experimental and theoretical findings resulted in a scheme depicting the link between the structure and optical properties of the liganded AuNCs (Figure 3). In that scheme (upper panel), the AuNC is presented as a multi-shell system that consists of three components: a metal core, the metal–ligand interface with staple motifs, and surface ligands. Surface thiolate ligands (SRs) do not just passivate the gold core but build staples or semi-rings, Au(SR)_2_ (-RS-Au-RS) or Au_2_(SR)_3_ (-RS-Au-S(R)-Au-SR-), that bind to the core surface and serve as its protection. The existence of three shells enables a ligand-to-metal core charge transfer (LMCT) or ligand-to-metal metal charge transfer (LMMCT). In both cases, communication can occur either through direct bonding or through the donation of electron-rich ligand groups. NIR and visible absorption of AuNCs is always a consequence of charge transfer, which occurs either due to metal–metal electron transitions or via LMCT and LMMCT (Figure 3, bottom panel). For instance, Zhou and coworkers revealed that the visible and NIR emissions of Au_25_ NCs originate from the surface state and Au_13_ core state, respectively [165]. 

#### 2.4.2. Quantum Nanoclusters as Non-Linear Optical Probes 

Liganded silver and gold NCs represent an emerging class of extremely interesting optical materials due to their remarkable NLO characteristics. SHG and/or TPA/TPEF processes can reach the highest corresponding cross-sections by a rational design of “ligand–core” templates and by controlling the NC size and/or asymmetry (Figure 4). A direct correlation between the structure and (multi)photonic properties of these nano-objects has been determined via experimental and theoretical investigations of the structure–property relationship (Figure 5). The TPA cross-section of liganded noble metal NCs is several orders of magnitude larger than that of commercially available dyes. These enhanced NLO properties are due to a subtle balance between resonance effects (position of transitions vs. laser excitation) and large transition dipole moments (due to ligand-to-core charge transfer character of excitations). On the other hand, the structure asymmetry (inherent to the metal core and/or brought by the asymmetric arrangement of surface ligands) in NCs can boost the SHG process (sensitive to non-centrosymmetric systems) [166], as evidenced by the enhanced NLO properties of Au_15_SG_13_ [167]. 

#### 2.4.3. Functionalization of Quantum Nanoclusters 

Functionalization is of crucial importance for creating tumor-biomarker-specific NCs with optimal bio-compatible optical properties. One of the great advantages of NCs is their suitability for efficient and controlled functionalization. Small organic molecules and biomolecules are commonly added to the NC surface as fluorophores to shift the optical properties of NCs towards the NIR region. Drugs, photosensitizers, or radiosensitizers can be used for functionalization for cancer therapy or as targeting molecules that interact specifically with receptors overexpressed at the surface of tumor cells. Such interfacial engineering of AuNCs for biomedical applications has been recently reviewed by Xie’s group [168]. One efficient approach to functionalization is a direct synthesis of AuNCs, with the molecule of interest containing a terminal thiol group that can bind to the metal surface. For instance, Le Guevel et al. [169] started with zwitterionic sulfobetaine-stabilized AuNCs that have the capacity to accumulate in brain tumors. To further improve the tumor uptake of these AuNCs, they functionalized them with arginine, and the resulting AuSG-2Arg exhibited rapid accumulation in cancer cells, thus being potentially interesting for radiotherapy enhancement [170]. A second approach is based on the post-functionalization of NCs, whereby click chemistry and succinimidyl ester reactions were used to covalently bind molecules of interest to the protective ligand [125].

A third approach, the ligand exchange strategy, is based on the replacement of the preexisting structural ligand with a specializing ligand containing a thiol group. The ligand exchange strategy is possible due to the unique structural features of the NCs, which can be prepared with atomic precision. This is in stark contrast to larger NPs, for which the control of surface functionalization is not possible. In addition to adding specific recognition properties to the NC, introducing a controlled number of functional ligand molecules by a ligand exchange strategy can also boost NCs’ NLO properties. Indeed, this ligand exchange will induce symmetry breaking in NCs, leading to efficient second-order nonlinear scattering, in particular for SHG signals, as demonstrated by Verbiest and colleagues [171]. Ligand shell engineering through ligand exchange can also increase the stability of metal NCs’ surface and lead to rigidification effects, enhancing their fluorescence properties [172,173]. The introduction of functional ligands through a ligand exchange strategy may also enhance their non-linear photoluminescence through a subtle interplay of metal–ligand interaction.

An example of a successfully applied ligand exchange strategy to introduce a specific recognition moiety into NCs is a recent generation of thiolated aminooxy-functionalized AuNCs, which can interact with protein carbonyls and be detected using optical methods (Figure 6) [167]. Au_15_SG_13_ NCs were readily functionalized by one or several thiolated aminooxy molecules (3-(aminooxy)-1-propanethiol) via a ligand exchange procedure. The as-prepared functionalized aminooxy-Au15 NCs were reacted with carbonylated proteins. The targeted carbonylated proteins were then detected either by one-photon fluorescence (with a fluorescence scanner) or by TPEF (with a multi-photon confocal microscope). Protein carbonylation at the molecular level on model lysozyme was validated by computation chemistry to better evaluate the nature of binding between the NCs and the protein carbonyls (see Figure 4). Altogether, this is proof of principle that functionalized liganded AuNCs can serve for the detection of carbonylation sites and might be more efficient than organic dyes. Such rational design of novel functional bi-thiolate-protected metal NCs with controllable surface chemistry could pave the way towards many similar practical applications, particularly in the molecular diagnostics of cancer.

#### 2.4.4. Application of the AuNCs in Cancer Diagnostics

To date, numerous AuNCs have been developed for the detection of tumor biomarkers such as the folate receptor, calreticulin, citrate, or the LHRH receptor (see Table 2). While many of them have been studied in cell lines, there are examples where they have been tested in liquid biopsies such as human urine and even in vivo in mouse models of cancer. For instance, Cys-AuNCs have been used to indirectly quantify the amount of citrate—a biomarker of early stages of prostate cancer—in a colorimetric assay applied in human urine [97]. This assay is based on a citrate-mediated inhibition of the intrinsic peroxidase-mimetic activity of Cys-AuNCs. Another example is an AuNC liganded with a tumor-targeting LHRH peptide and labeled with iodine-124, which has been used as a PET tracer for lung cancer in xenografted mice [74]. Even though AuNCs are still insufficiently explored, these and other similar studies pave the way for their future application in the clinical setting.

## 3. Conclusions—Perspectives 

Various nanomaterials have been developed as tools of precision medicine for the detection of tumor biomarkers. Liganded metal nanomaterials such as NPs and NCs are structurally highly versatile, thus providing numerous opportunities for specific applications. They can be adapted to detect different biomarkers and for visualization by different imaging techniques through the selection of suitable specializing (biomarker recognition moieties) and structural ligands, respectively. Given the increasing number of newly developed nanomaterials and options for their detection, the choice of a specific nanomaterial in a desired application becomes a challenge. In this review, we have discussed atomically precise ligand-protected noble metal NCs and their properties relevant for tumor biomarker detection (summarized in Figure 7). They are characterized by non-linear optical properties that are compatible with biological samples and allow for deep-tissue imaging. Moreover, their functionalization can be precisely controlled. Finally, they can be rapidly eliminated from the body via the renal clearance system. Together, these considerations emphasize the value of NCs as tools in the molecular diagnostics of cancer. 

## Figures and Tables

**Figure 1 cancers-13-04206-f001:**
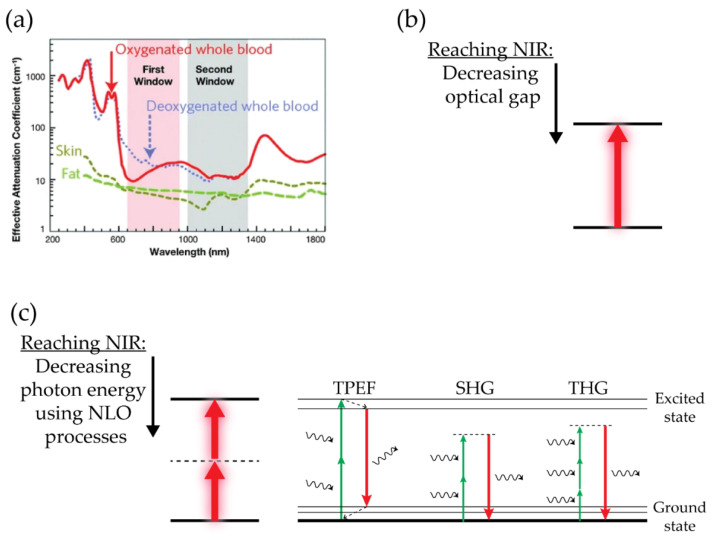
Optical imaging in biological samples. (**a**) Tissues such as skin, fat, and blood (oxygenated and deoxygenated) attenuate light in a wavelength-dependent manner due to absorption and scattering. Attenuation is the lowest in the first (NIR-I, 700–900 nm, shaded in pink) or second near-infrared window (NIR-II, 1000–1700 nm, shaded in grey). Image adapted from [112] with permission from Springer Nature. (**b**,**c**) Strategies for reaching the NIR-I and NIR-II windows: using chromophores with small optical gaps (**b**) or applying NLO techniques (SHG, THG, and TPEF) (**c**). Images adapted from [113].

**Figure 2 cancers-13-04206-f002:**
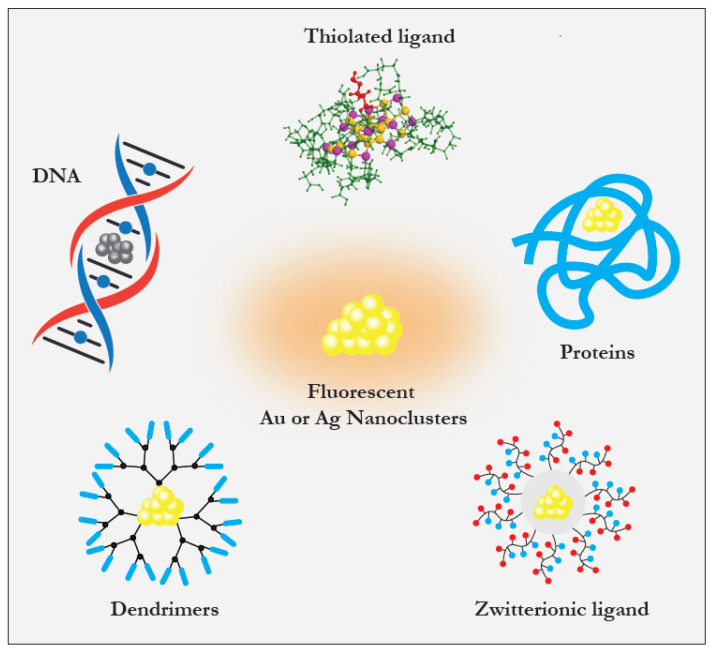
Metal nanoclusters protected with different scaffolds.

**Figure 3 cancers-13-04206-f003:**
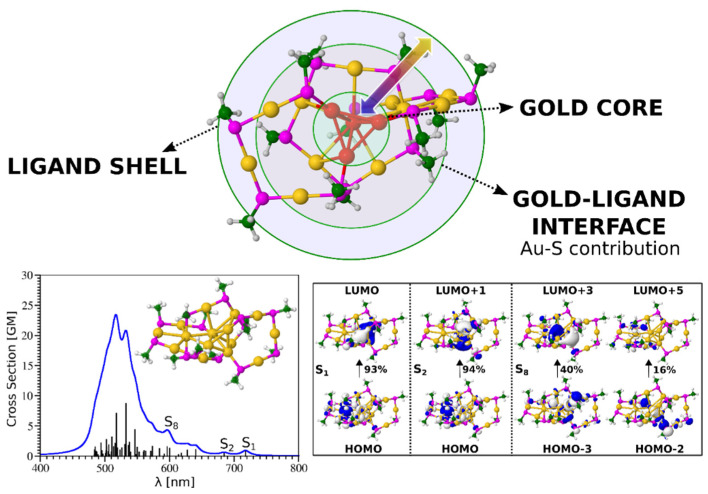
Diagram showing the view of liganded gold nanoclusters as multi-shell system on an example of Au_15_(SCH_3_)_13_ (**upper panel**). TDDFT two-photon absorption spectrum (**bottom left**) and molecular orbitals (**bottom right**) involved in transitions for an Au_15_(SCH_3_)_13_ nanocluster. The upper scheme was adapted from [45] (p. 18). For theoretical methods used for the **bottom panel**, see [113].

**Figure 4 cancers-13-04206-f004:**
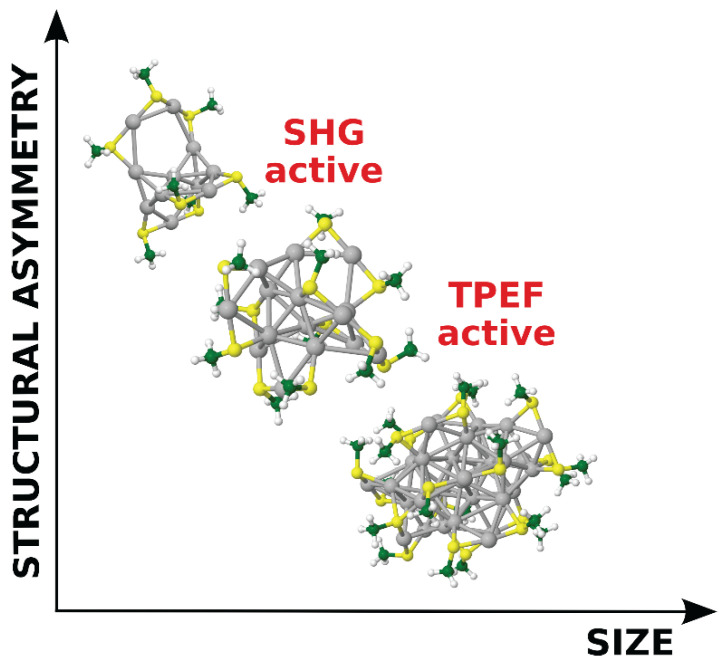
NLO properties of liganded metal nanoclusters as a function of size and asymmetry (Ag_11_, Ag_15_, and Ag_32_ are shown as examples).

**Figure 5 cancers-13-04206-f005:**
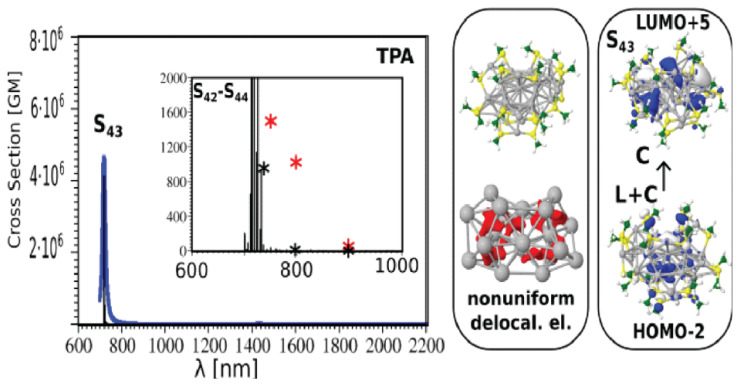
TDDFT TPA absorption spectrum (and experimental values in insets, red crosses) and molecular orbitals involved in transitions for Ag_31_(SCH3)_19_ nanoclusters (right side). Giant TPA cross-sections have been reported.

**Figure 6 cancers-13-04206-f006:**
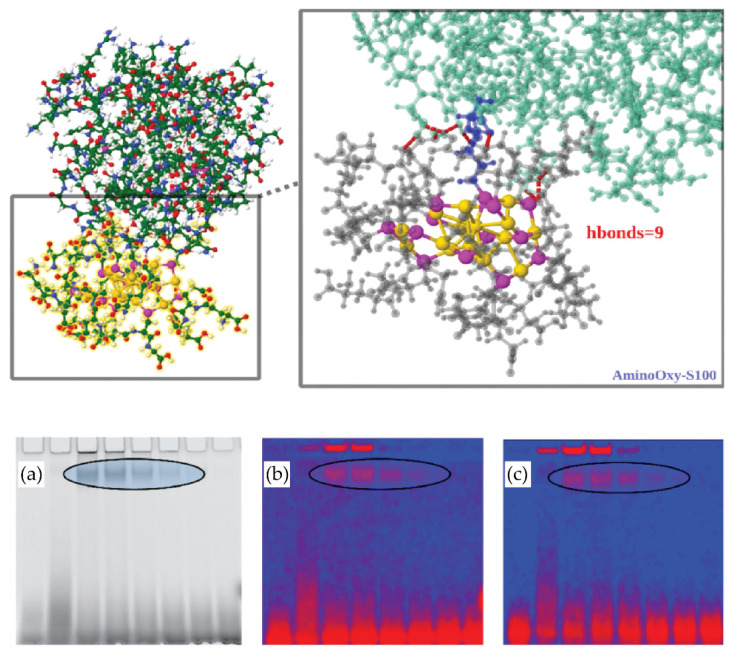
Detection of protein carbonyls using aminooxy-liganded AuNCs (top panel). Quantum mechanics/molecular mechanics (QM/MM) (two-layer ONIOM [174]) interacting with lysozyme. The interface between liganded cluster Au_15_(SG)_12_(3-(aminooxy)-1-propanethiol)1 and protein is enlarged (right side), illustrating the aminooxy-serine oxyme bond and the hydrogen-bonding network (SG—grey; Au—gold; S—magenta; lysozyme—green; oxyme bond—blue; h-bonds—dotted in red). Figure adapted from [167] (bottom panel). Au15-targeted carbonylated proteins detected on gels by fluorescence scanner (**a**) and multiphoton confocal imaging (**b**,**c**). Laser excitation was at 780 nm, and emitted photons were detected with (**b**) visible range (350–700 nm) and (**c**) IR range (>850 nm). Ellipses: Fixed concentration of 500 µM Au_15_(SG)_12_(3-(aminooxy)-1-propanethiol)1 was incubated with a decreasing range of concentrations of the lysozyme (50–1 µM, corresponding to 5–0.1 µg protein loaded in the gel).

**Figure 7 cancers-13-04206-f007:**
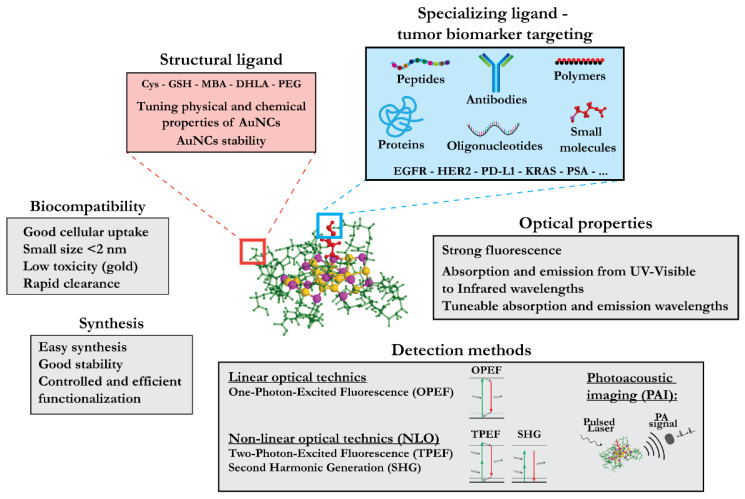
Summarized properties of liganded AuNCs as cancer diagnostics agents.

**Table 1 cancers-13-04206-t001:** Techniques used for cancer diagnosis.

Diagnostic Method	Principle	Technique	Detection Agent/Visualization
Imaging	Function-based	PET	Radioactive tracers that produce positrons/Scanner
	Function-based	SPECT	Radioactive tracers that emit gamma rays/Scanner
	Anatomy-based	MRI	Magnet, radiofrequency/Scanner
	Anatomy-based	CT	X-ray/Scanner
	Hybrid (anatomy- and function-based)	PET/CT, PET/MRI, SPECT/CT	Combination of radioactive tracer and imaging modality
	Optical	PA, SERS	Luminescent probe/Scanner
Molecular diagnostics	Gene amplification	PCR	DNA sequencing
	Cytogenetic analysis—hybridization of nucleic acids in cells/tissues	FISH	Fluorescent labels/Fluorescent microscopy
	Hybridization of nucleic acids in microplates	DNA microarrays	Labels/Microscopy
	Immunoassay for protein detection	ELISA	Luminescent probe/Plate reader

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
