# Peer review of "Nanotechnology in Tumor Biomarker Detection: The Potential of Liganded Nanoclusters as Nonlinear Optical Contrast Agents for Molecular Diagnostics of Cancer"

_cancers, 2021, doi:10.3390/cancers13164206_

Round 1

Reviewer 1 Report

Your review article is a very good piece of work and explain very well the evolution of cancer diagnostics, specially focusing in the application of metallic nanomaterials to this field. I found the manuscript well written, well organized and complete. However there are a high amount of technical information about imaging, optical techniques and the physical properties of the nanoclusters, that I‘m afraid do not fit in the scope of this journal. Therefore, to make the text more understandable for a higher number of readers with a very different knowledge backgrounds, I recommend

  1. To add summary tables to facilitate the reading in section 1 and 2 to compare the different techniques presented in the text.
  2. In section 2 add a little introduction (just few lines) connecting the part 1 of the text, with part 2, better justifying why you focus in the use of quantum nanoclusters and their importance in the case of cancer.
  3. Add more practical examples of the application of quantum nanocluster in the cancer field.

Reviewer 2 Report

The manuscript by Combes et al. titled ´´Nanotechnology in Tumor Biomarker Detection: Potential of Liganded Nanoclusters as Nonlinear Optical Contrast Agents for Molecular Diagnostics of Cancer’’ attempt to highlight the use of nanotechnology potential for molecular diagnostic of cancer. Moreover, in this work, the authors focus on the versatility of nanomaterials to decorate with a ligand at nanoparticles surface which increases their specificity. This approach is relevant due to tumour heterogeneity which has implications in the diagnosis and cancer treatment, contributing to the knowledge for precision medicine. Although this manuscript is well written, organized and properly discuss the novel character of this work is minor.

In chapter 1 the authors refer that early cancer diagnosis decreases the mortality rates but also reduces the co-morbidity rates and that information should be included once his two-factor decrease cancer burden, beyond human factor, also has an economic impact. In addition, this chapter should include some oncology epidemiology information.

Furthermore, in table 1, the polydispersity index should be included, a measure of the heterogeneity of the sample based on size. The agglomeration process could occur during NP formulation or functionalization process. The importance of this parameter on the physical-chemical characterization of nanomaterials should be at least discussed.

Considering the importance of early diagnosis, how these novel technologies could detect tumour biomarkers in circulation? If the authors considering that this subject is relevant, they may include some information about it.

Also, on page 14 section 2.3.3. Biodistribution, the authors refer ´´protein corona model does not seem to apply to the NCs’’, which is not completely true once so far have only been a few attempts to address this important issue, that affect influences NCs biological behaviour as biodistribution, cellular uptake and cytotoxicity.

Given the clinical importance of this work, the authors could include some information about the nanoparticles for molecular diagnostic of cancer already in clinical trials and their impact.
